# Screening for insulin-independent pathways that modulate glucose homeostasis identifies androgen receptor antagonists

Sri Teja Mullapudi[1], Christian SM Helker[1†], Giulia LM Boezio[1], Hans-Martin Maischein[1], Anna M Sokol[2], Stefan Guenther[3], Hiroki Matsuda[1‡], Stefan Kubicek[4], Johannes Graumann[2,5], Yu Hsuan Carol Yang[1], Didier YR Stainier[1]*

[1]Department of Developmental Genetics, Max Planck Institute for Heart and Lung Research, Bad Nauheim, Germany; [2]Biomolecular Mass Spectrometry, Max Planck Institute for Heart and Lung Research, Bad Nauheim, Germany; [3]ECCPS Bioinformatics and Deep Sequencing Platform, Max Planck Institute for Heart and Lung Research, Bad Nauheim, Germany; [4]CeMM Research Center for Molecular Medicine of the Austrian Academy of Sciences, Vienna, Austria; [5]German Centre for Cardiovascular Research, Berlin, Germany

*For correspondence:
Didier.Stainier@mpi-bn.mpg.de

Present address: †Faculty of Biology, Cell Signaling and Dynamics, Philipps-University Marburg, Marburg, Germany; ‡Department of Biomedical Sciences, College of Life Sciences, Ritsumeikan University, Kusatsu, Japan

**Abstract** Pathways modulating glucose homeostasis independently of insulin would open new avenues to combat insulin resistance and diabetes. Here, we report the establishment, characterization, and use of a vertebrate 'insulin-free' model to identify insulin-independent modulators of glucose metabolism. *insulin* knockout zebrafish recapitulate core characteristics of diabetes and survive only up to larval stages. Utilizing a highly efficient endoderm transplant technique, we generated viable chimeric adults that provide the large numbers of *insulin* mutant larvae required for our screening platform. Using glucose as a disease-relevant readout, we screened 2233 molecules and identified three that consistently reduced glucose levels in *insulin* mutants. Most significantly, we uncovered an insulin-independent beneficial role for androgen receptor antagonism in hyperglycemia, mostly by reducing fasting glucose levels. Our study proposes therapeutic roles for androgen signaling in diabetes and, more broadly, offers a novel in vivo model for rapid screening and decoupling of insulin-dependent and -independent mechanisms.
DOI: https://doi.org/10.7554/eLife.42209.001

## Introduction

Characterized by the inability to control blood glucose levels, diabetes is a metabolic disease of major socio-economic concern. Blood glucose levels are regulated by multiple tissues including the pancreas, muscle, liver, adipocytes, gut and kidney (*Defronzo, 2009*). Signals from endocrine hormones are integrated by each tissue to effectively maintain glucose homeostasis, and aberrations in this interplay underlie the pathogenesis of diabetes. Currently, seven classes of antidiabetic drugs exist, of which only three function without increasing circulating insulin levels and only one that definitively functions independently of insulin (*Chaudhury et al., 2017*). Restoring normoglycemia independently of insulin secretion or action could delay disease progression as an improved glycemic status can restore β-cell mass and function (*Wang et al., 2014*). Lower dependence on insulin-stimulating therapies can also prevent hyperinsulinemia-driven insulin resistance (*Shanik et al., 2008*) and obesity (*Mehran et al., 2012*). In contrast to insulin stimulators, Biguanides (e.g.,

**eLife digest** Diabetes is a disease that affects the ability of the body to control the level of sugar in the blood. Individuals with diabetes are unable to make a hormone called insulin – which normally stimulates certain cells to absorb sugar from the blood – or their cells are less able to respond to this hormone. Most treatments for diabetes involve replacing the lost insulin or boosting the hormone's activity in the body. However, these treatments can also cause individuals to gain weight or become more resistant to insulin, making it harder to control blood sugar levels.

In addition to insulin, several other factors regulate the levels of sugar in the blood and some of them may operate independently of insulin. However, little is known about such factors because it is impractical to carry out large-scale screens to identify drugs that target them in humans or mice, which are often used as experimental models for human biology.

To overcome this challenge, Mullapudi et al. turned to another animal known as the zebrafish and generated mutant fish that lack insulin. The mutant zebrafish had similar problems with regulating sugar levels as those observed in humans and mice with diabetes. This observation suggests that insulin is just as important in zebrafish as it is in humans and other mammals.

The mutant zebrafish did not survive into adulthood, and so Mullapudi et al. transplanted healthy tissue into the zebrafish to allow them to produce enough insulin to survive. These adult zebrafish produced many offspring that still carried the insulin mutation. Mullapudi et al. used these mutant offspring to screen over 2,000 drugs for their ability to decrease blood sugar levels in the absence of insulin. The screen identified three promising candidate drugs, including a molecule that interferes with a receptor for a signal known as androgen.

These findings will help researchers investigate new ways to treat diabetes. In the future, the screening approach developed by Mullapudi et al. could be adapted to search for new drugs to treat other human metabolic conditions.

DOI: https://doi.org/10.7554/eLife.42209.002

Metformin) and Thiazolidinediones (e.g., Pioglitazone) are effective antidiabetic agents that primarily sensitize tissues to insulin (reviewed by (*Soccio et al., 2014*; *Rena et al., 2017*)). Likewise, sodium-glucose transporter two inhibitors (e.g., Dapagliflozin) have a complementary mechanism of reducing glucose reabsorption in the kidney (*Bailey et al., 2013*). Increasing evidence points to additional molecular pathways that can improve metabolic homeostasis independently of insulin, for instance, using leptin therapy (*Neumann et al., 2016*) or exercise (*Stanford and Goodyear, 2014*). Interestingly, currently prescribed drugs were discovered from their historical use in herbal medicine (*Ehrenkranz et al., 2005*; *Bailey, 2017*) or from screens directed against hyperlipidemia (*Fujita et al., 1983*). However, so far, an unbiased search for insulin-independent pathways controlling glucose metabolism has remained elusive, primarily due to the lack of a disease-relevant animal model for rapid screening. Due to its high fecundity and amenability to chemical screening, the zebrafish serves as an excellent platform to study diabetes, and it has been successfully used to study β-cell mass and activity, as well as glucose metabolism (*Andersson et al., 2012*; *Gut et al., 2013*; *Tsuji et al., 2014*; *Nath et al., 2015*; *Li et al., 2016*; *White et al., 2016*; *Gut et al., 2017*; *Matsuda et al., 2018*). Here, using the zebrafish model, we generated an innovative drug discovery strategy, screened chemical libraries and specifically identified insulin-independent effects of androgen signaling on glucose homeostasis.

## Results and discussion

### *insulin* is crucial for zebrafish metabolic homeostasis already at larval stages

Insulin plays a central role in glucose homeostasis by increasing glucose uptake in peripheral tissues, promoting glycogenesis in the liver and decreasing glucose production by inhibiting glucagon secretion (*Aronoff et al., 2004*). We generated zebrafish devoid of insulin signaling and determined the degree to which these mutants recapitulate core features of diabetic metabolism observed in mammals. The zebrafish genome contains two insulin genes – *insulin (ins)* and *insulinb (insb)*. Using

CRISPR/Cas9 mutagenesis, we generated a 16 bp deletion allele of *ins* (*Figure 1A*) and a 10 bp insertion allele of *insb*. Although *ins* and *insb* mutant embryos appear morphologically unaffected (*Figure 1—figure supplement 1A*), Insulin was entirely absent in pancreatic islets of *ins* mutants (*Figure 1B*), whereas there was no observable change in *insb* mutant islets (*Figure 1—figure supplement 1B and C*). *ins* mutants exhibit a drastic increase in total glucose levels (up to 10-fold), measured from 1 to 6 days post fertilization (dpf) (*Figure 1C*). Additionally, staining for lipid content using Nile Red revealed large unused yolk reserves (*Figure 1D*), suggesting defects in lipid absorption and processing. Due to a combination of these metabolic defects, *ins* mutants do not survive beyond 12 dpf (*Figure 1E*). Moreover, although 3 month old (adult) *ins +/-* animals are normoglycemic (*Figure 1—figure supplement 1D*), 50 dpf (juvenile) *ins +/-* animals are noticeably smaller (*Figure 1—figure supplement 1E*), consistent with a role for Insulin in growth control (*Nakae et al., 2001*). *insb* mutants, on the other hand, are viable and fertile. During WT development, *insb* expression is minimal beyond 48 hpf (*Papasani et al., 2006*; *White et al., 2017*) (*Figure 1—figure supplement 1F*), and it is undetectable in adult β-cells (*Tarifeño-Saldivia et al., 2017*). To assess whether it is capable of function, we overexpressed *insb* under the *ins* promoter. Under the hyperglycemic conditions resulting from morpholino (MO)-mediated *ins* knockdown, *insb* overexpression successfully lowered glucose levels, thus indicating that *insb* is functional (*Figure 1—figure supplement 1G*). However, due to the post-embryonic expression of *ins*, survival and metabolic homeostasis in zebrafish depends primarily on *ins*. This predominant role of *ins* distinguishes the zebrafish insulins from the redundant metabolic roles of mouse *Ins1* and *Ins2* (*Duvillié et al., 1997*). To further explore the nature of the metabolic defects in zebrafish *ins* mutants, we probed the proteome of 108 hpf *ins* mutants compared to their WT siblings and assessed this dataset relative to seven other proteomes from diabetic tissues of rodent or human origin (*Hwang et al., 2010*; *Mullen and Ohlendieck, 2010*; *Giebelstein et al., 2012*; *Valle et al., 2012*); S.-J. *Kim et al., 2014a*; *Capuani et al., 2015*; *Braga et al., 2016*; *Zabielski et al., 2016*). Strikingly, pathways like gluconeogenesis, mitochondrial dysfunction, sirtuin signaling, and oxidative phosphorylation, which were affected in diabetic conditions across these studies, were similarly disrupted in zebrafish *ins* mutants (*Figure 1F* and *supplementary file 1*). Together, these findings indicate that zebrafish *ins* is crucial for metabolic homeostasis and survival, and that its absence causes core features of diabetic metabolism already at larval stages.

## Highly efficient endoderm transplant technique rescues *ins* mutants to adulthood

Screening of small molecules in *ins* mutants requires large numbers of mutant animals. However, the early lethality of *ins* mutants did not allow the generation of adult animals that can be incrossed. To overcome this obstacle, we used an efficient endoderm induction (*Kikuchi et al., 2001*) and transplantation technique (*Stafford et al., 2006*) (*Figure 2A*) to selectively modify endodermal tissues without altering the germline. *Tg(ins:DsRed); ins +/+* embryos were injected with *sox32* mRNA at the one-cell stage, conferring an endodermal fate on all cells. Between 3 to 4 hpf, cells were transplanted from these embryos to the mesendoderm of similarly staged embryos from *Tg(ins:RasGFP); ins +/-* incrosses. This transplantation procedure was remarkably efficient at contributing to host endoderm (*Figure 2—figure supplement 1A–A''*), and the pancreatic islet of nearly every host embryo contained both donor derived (i.e., *ins +/+*) as well as host β-cells (*Figure 2B–B''*). These chimeric animals were raised to adulthood and genotyping (*Figure 2—figure supplement 1B–C*) revealed a near Mendelian ratio of mutant animals (*Figure 2C*). In summary, these chimeric animals contain *ins +/+* endodermal tissues but retain an *ins -/-* germline, thereby allowing an all-mutant progeny to be obtained by incrossing.

## Small molecule screen in *ins* mutants reveals insulin-independent modulators of glucose metabolism

With the ability to obtain large numbers of *ins* mutant embryos, we next aimed to analyze the effect of known glucose homeostasis modulators and also to screen for novel ones. We tested the effects of molecules that have been proposed to help normalize glucose levels in an insulin-independent manner as well as others that do so in an insulin-dependent manner. Anti-diabetics such as metformin, pioglitazone and dapagliflozin, as well as the Lyn kinase activator MLR1023 (*Saporito et al.,*

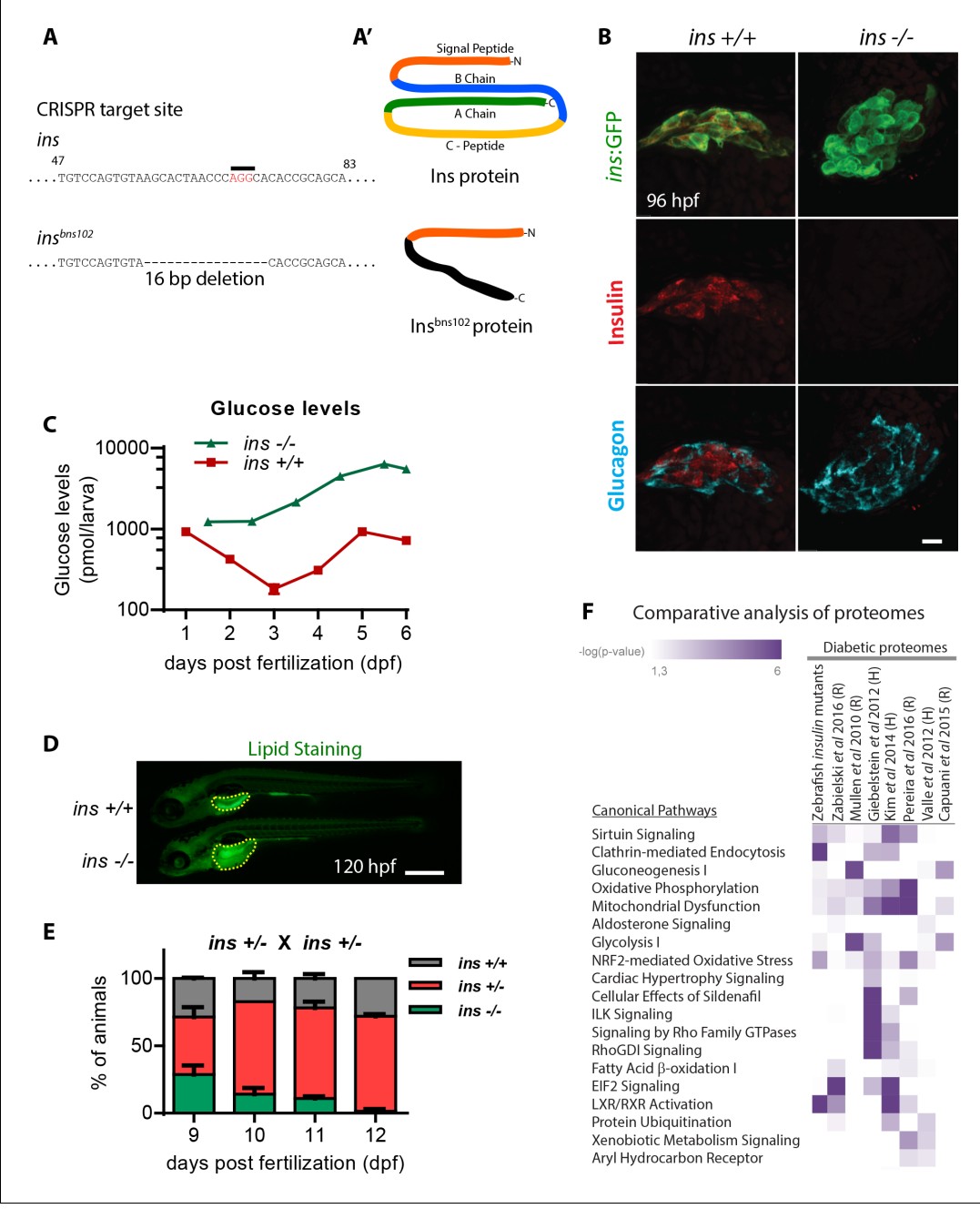

**Figure 1.** *insulin* is crucial for zebrafish metabolic homeostasis. (**A**) CRISPR target site in the *insulin* gene, with PAM sequence highlighted, and the resulting 16 bp deletion allele (below). (**A'**) Schematic of wild-type Insulin protein and the predicted mutant protein which contains novel sequence (black). (**B**) Confocal projection images of the pancreatic islet in 96 hpf *Tg(ins:GFP) ins +/+* and *ins -/-* animals immunostained for Insulin (red) and Glucagon (cyan). (**C**) Free glucose levels in wild-type and mutant animals from 1 to 6 dpf; mean ± SEM, n = 2–4 replicates. (**D**) Nile Red staining (green) for neutral lipids in 120 hpf wild-type (top) and mutant (bottom) larvae, with yolk lipid content outlined (yellow dots). (**E**) Genotype distribution from *ins* ± incross, calculated as the percentage of total animals at each stage; mean ± SEM, n = 32 animals at each stage. (**F**) Heat map of the proteomic signature of zebrafish *ins* mutants at 120 hpf compared to signatures from rodent (**R**) and human (**H**) diabetic proteome studies. Canonical pathways implicated in most studies are listed first. P-value cut-off set at <0.05. Scale bars: 10 μm (**B**), 500 μm (**D**).

DOI: https://doi.org/10.7554/eLife.42209.003

The following figure supplement is available for figure 1:

*Figure 1 continued on next page*

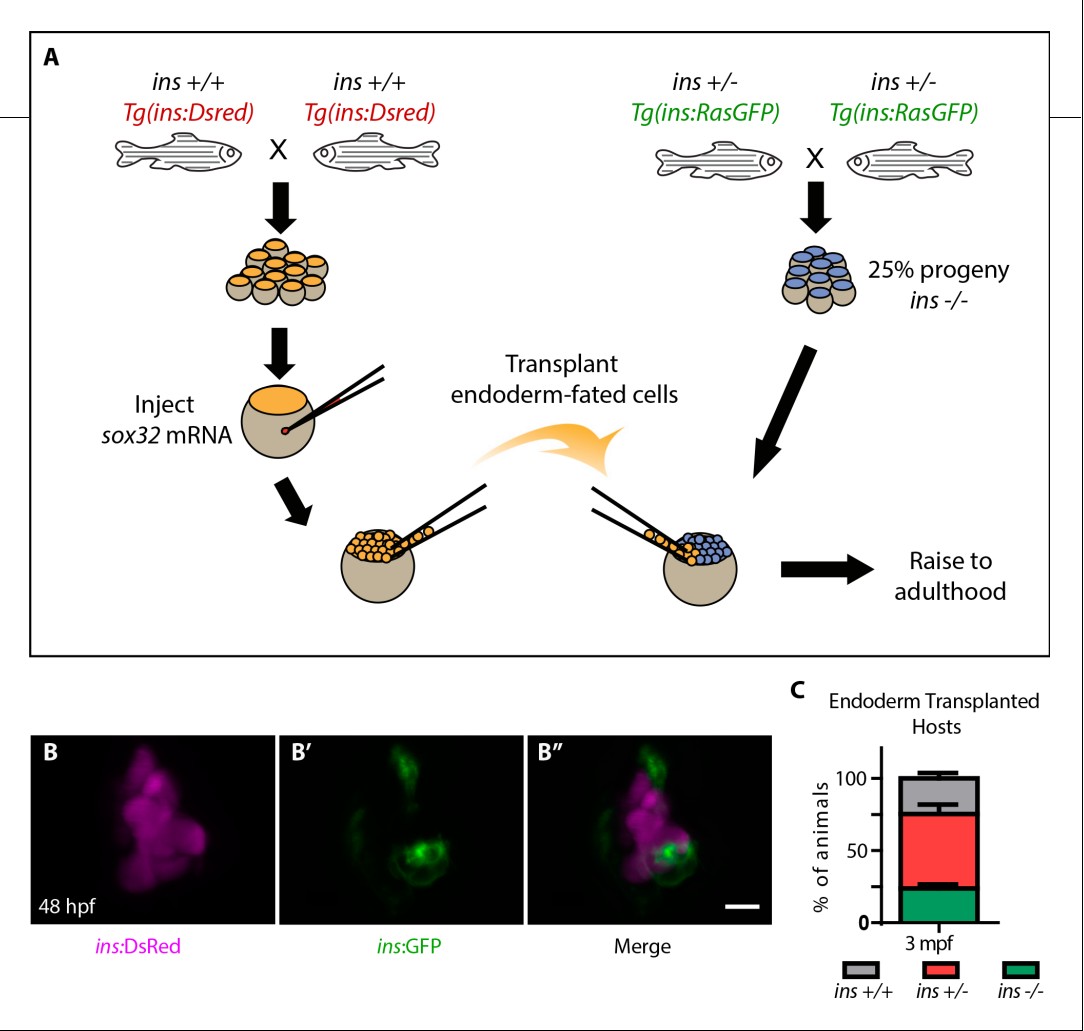

**Figure 2.** Highly efficient endoderm transplant technique rescues *ins* mutants to adulthood. (**A**) Schematic depicting the endoderm transplantation protocol; *sox32* mRNA-injected *ins +/+* donor cells (orange) were transplanted into host embryos (blue) to form chimeric animals. (**B–B''**) Confocal projection images of the pancreatic islet of a 48 hpf chimeric animal showing β-cells from the host (green, (**B'**) and the transplanted *ins +/+* cells (magenta, (**B**). (**C**) Genotype distribution in the raised three mpf chimeric animals, determined by genotyping fin tissue; mean ± SEM, n = 3 transplant experiments, 18–32 animals per experiment. Scale bar: 10 μm.
DOI: https://doi.org/10.7554/eLife.42209.005
The following figure supplement is available for figure 2:

**Figure supplement 1.** *sox32* mRNA-injected cells contribute to host endoderm upon transplantation.
DOI: https://doi.org/10.7554/eLife.42209.006

*2012*), were tested. We also tested fraxidin, identified in a screen for molecules that increase glucose uptake in zebrafish (*Lee et al., 2013*). Surprisingly, metformin and MLR1023 exhibited no glucose-lowering effect in *ins* mutants, suggesting that they act more as sensitizers of insulin signaling rather than independently of insulin (*Figure 3A*). In *ins* mutants, Pck1 levels are higher compared to wild types (*Supplementary file 1*), suggesting increased gluconeogenesis in the absence of the inhibitory action of insulin. This observation is also supported by the drastic reduction of glucose levels in *ins* mutants treated with the Pck1 inhibitor, 3 MPA (*Figure 3A*). Metformin has well-known abilities to reduce hepatic gluconeogenesis, and it also reduces glucose levels in isoprenaline-treated wild-type zebrafish (*Gut et al., 2013*). However, metformin's inability to reduce glucose levels in zebrafish *ins* mutants reveals metformin's dependence on insulin signaling for its action. MLR1023's glucose level lowering effects have been proposed to be insulin-dependent (*Ochman et al., 2012*),

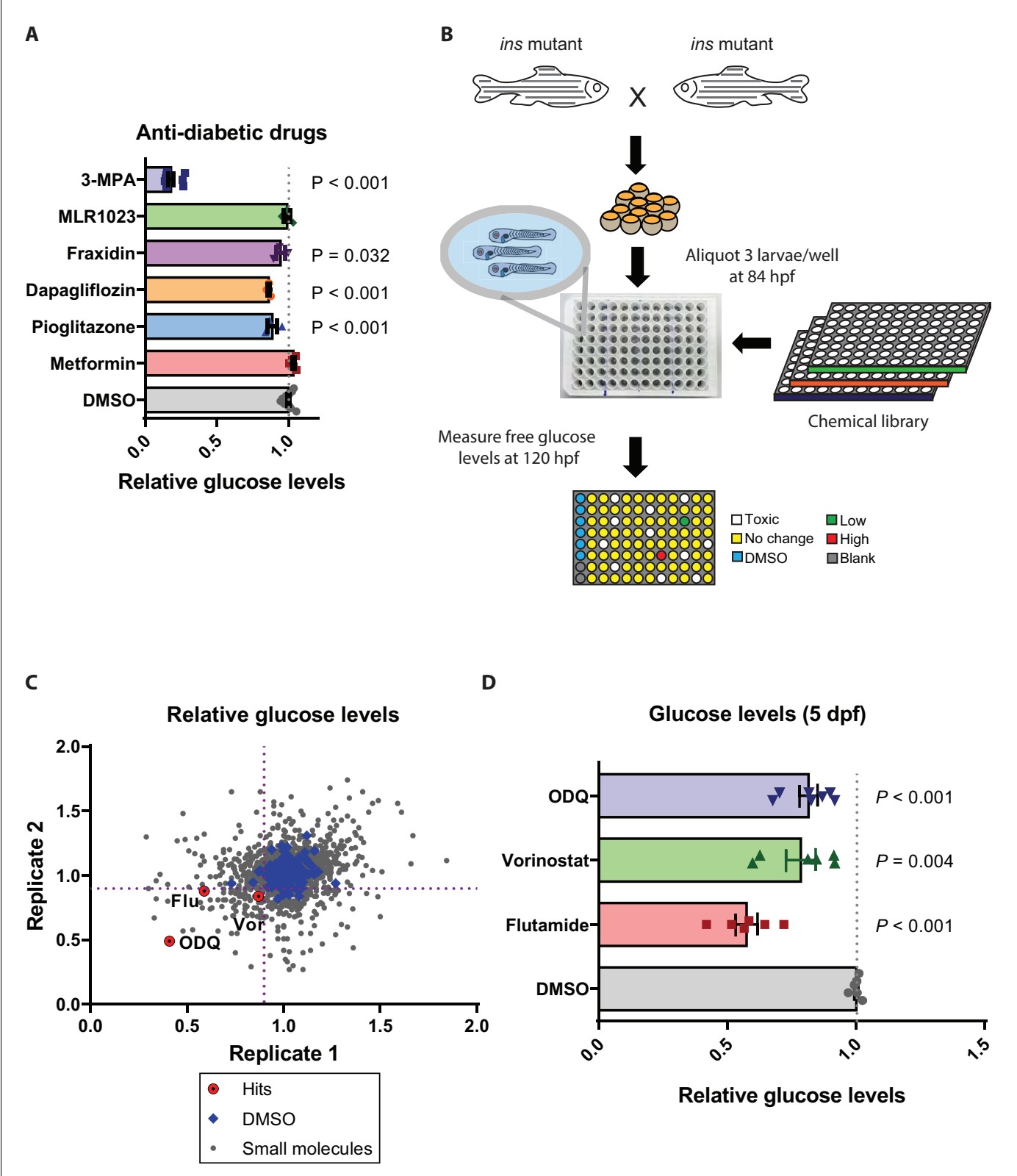

**Figure 3.** Small molecule screen in *ins* mutants reveals insulin-independent modulators of glucose metabolism. (**A**) Relative glucose levels in 120 hpf *ins* mutant larvae after 36 hr of treatment with anti-diabetic drugs (dapagliflozin, pioglitazone, metformin) or reported insulin mimetics (MLR1023, Fraxidin) or the Pck1 inhibitor, 3 MPA; mean ± SEM, n = 3–7 replicates. (**B**) Schematic representation of the screening pipeline: *ins* mutant larvae were treated with small molecules starting at 84 hpf and free glucose levels measured at 120 hpf. (**C**) Scatter plot showing relative change in glucose levels upon
*Figure 3 continued on next page*

*Figure 3 continued*

treatment with 2233 small molecules. X and Y axes represent two replicates performed for each drug, with the dotted purple lines marking 0.9 on each axis. 72 molecules satisfied the pre-specified cut-off. (D) Relative glucose levels at 120 hpf upon treatment of *ins* mutants with the three hits – ODQ, Vorinostat, and Flutamide; mean ± SEM, n = 6–7 replicates. *P* values from t-tests.

DOI: https://doi.org/10.7554/eLife.42209.007

The following figure supplement is available for figure 3:

**Figure supplement 1.** A 96-well plate-adapted protocol to measure glucose levels is suitable for small molecule screening.

DOI: https://doi.org/10.7554/eLife.42209.008

and thus, its inability to lower glucose levels in zebrafish *ins* mutants further suggests the lack of any insulin signaling in these animals. On the other hand, fraxidin, dapagliflozin, and pioglitazone reduced glucose levels by 5, 12, and 11% respectively (*Figure 3A*), thus attributing part of their glucose lowering effect to an insulin-independent mechanism. The lack of adipose tissue (*Minchin and Rawls, 2017*) and the primitive nature of kidney (pronephros) function at these developmental stages (*Elmonem et al., 2018*) may result in an incomplete recapitulation of adipose signaling and renal function on glucose homeostasis (*Defronzo, 2009*). This limitation could also explain the small magnitude of glucose level reduction observed with the PPARγ agonist or SGLT2 inhibitor treatments in zebrafish *ins* mutants. Based on these data with known glucose level lowering drugs, we decided to screen chemical libraries to identify molecules that could reduce glucose levels by more than 10% in zebrafish *ins* mutants.

To rapidly measure free glucose levels in a 96-well plate format, we adapted a kit-based protocol that is sensitive to endogenous changes in larval glucose levels (*Figure 3—figure supplement 1A–C*), and established a screening pipeline (*Figure 3B*). We screened 2233 molecules in 2 replicates at 10 μM concentration and found three hits (*Figure 3C*) that reproducibly reduced glucose levels upon retesting with independent chemical stocks and the unmodified standard glucose measurement kit. These three hits - flutamide (androgen receptor antagonist), ODQ (soluble guanylyl cyclase (sGC) inhibitor (*Boulton et al., 1995*)) and vorinostat (broad HDAC inhibitor (*Finnin et al., 1999*)) were found, upon retesting multiple times, to reduce glucose levels by 40, 22% and 19%, respectively (*Figure 3D*, *Figure 3—figure supplement 1D*). sGC inhibition by ODQ has been previously reported to increase net hepatic glucose uptake and shift the balance towards glycogenesis (*An et al., 2010*). Contrary to our findings, clinical use of vorinostat has been associated with hyperglycemia as a side effect (*Mann et al., 2007*). This difference could be due to the broad nature of Vorinostat's HDAC inhibition properties including anti-proliferative effects (*Richon, 2006*), which are likely to affect developmental processes in zebrafish *ins* mutants. These three drugs did not prolong the survival of zebrafish *ins* mutants, likely because lowering glucose levels alone was not sufficient to normalize all the metabolic, growth, and differentiation processes (*Taniguchi et al., 2006*) dysregulated in these animals.

## Androgen receptor (AR) antagonism reduces glucose levels in hyperglycemic larval and adult animals

Given the strong reduction in glucose levels observed after flutamide treatments, we further tested the hypothesis that glucose levels in *ins* mutants were being reduced through androgen receptor antagonism. First, flutamide caused a dose-dependent decrease in glucose levels in *ins* mutants (*Figure 4—figure supplement 1A*). Second, we treated *ins* mutants with AR antagonists of two types: (i) steroidal (Cyproterone) and (ii) non-steroidal (nilutamide, hydroxyflutamide, bicalutamide, enzalutamide), and observed a consistent decrease in glucose levels across all treatments, albeit at varying efficiency (*Figure 4A*), possibly reflecting the different efficacy of these antagonists towards zebrafish AR (*Raynaud et al., 1979*; *Teutsch et al., 1994*; *Tran et al., 2009*). Finally, to modulate AR protein levels, we injected 1 ng of control or *ar* MO into one-cell stage embryos and observed a reduction of glucose levels in *ar* MO injected *ins* mutants (*Figure 4B*) but not in *ar* MO injected wild-type animals (*Figure 4C*). These data support a role, and a benefit, for antagonizing AR specifically in hyperglycemic conditions.

A number of mechanisms have been proposed to explain the predisposition of women with androgen excess to diabetes, including insulin resistance, visceral adiposity, and β-cell dysfunction

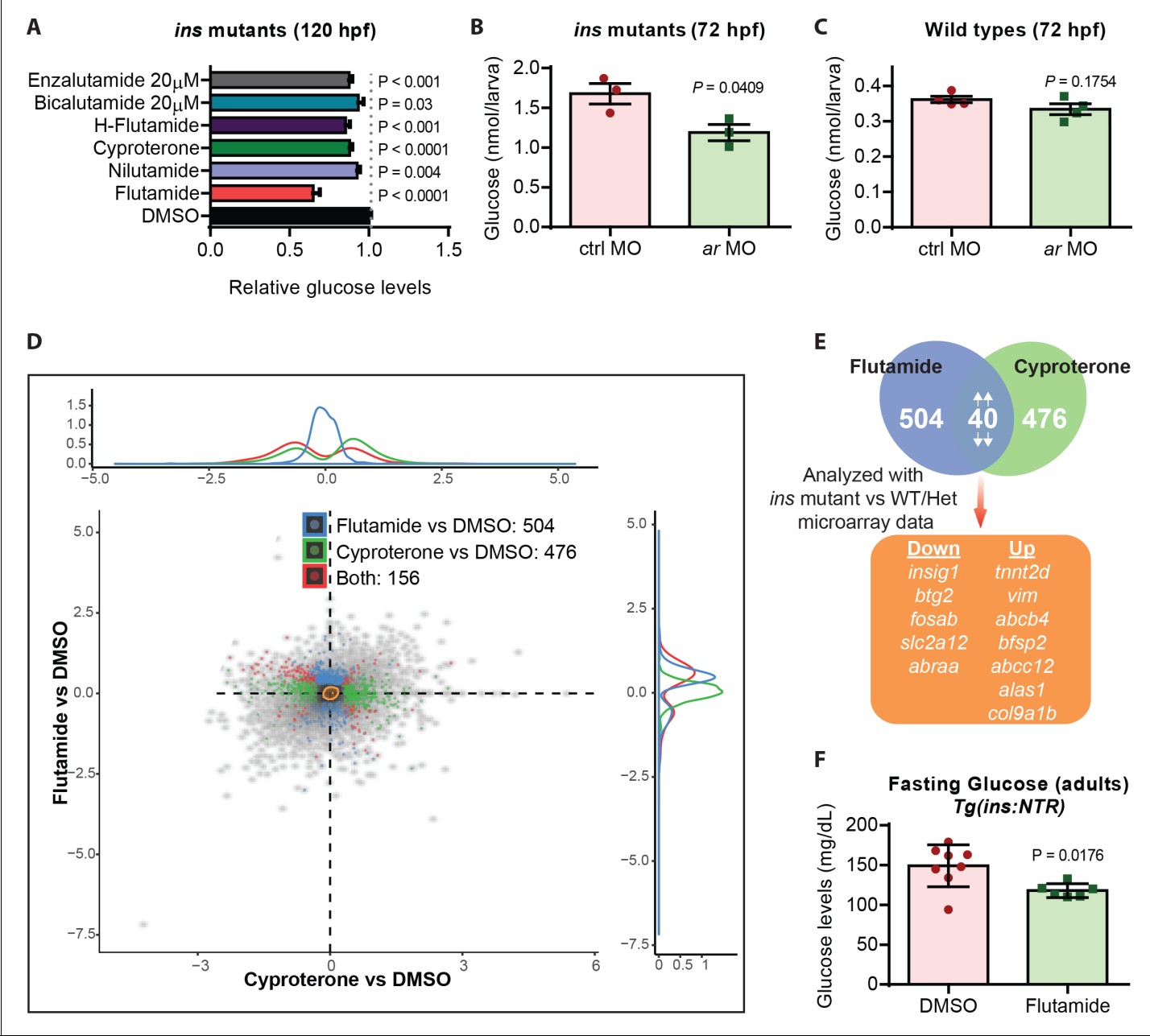

**Figure 4.** Androgen receptor (AR) antagonism reduces glucose levels in hyperglycemic larvae and adults. (A) Relative glucose levels in *ins* mutants at 120 hpf upon treatment with various AR antagonists starting at 84 hpf; mean ± SEM, n = 3–7 replicates. (B) Glucose levels in 72 hpf *ins* mutants after injection with 1 ng of ctrl or *ar* MO; mean ± SEM, n = 3 replicates. (C) Glucose levels in 72 hpf wild types after injection with 1 ng of ctrl or *ar* MO; mean ± SEM, n = 4 replicates. (D) RNA-seq analysis of 120 hpf *ins* mutant larvae treated with flutamide or cyproterone starting at 84 hpf, showing differentially expressed genes (DEGs) compared to DMSO-treated larvae in blue and green, respectively. Red dots indicate DEGs common to both treatments. (E) Workflow used to filter candidate genes: 40 DEGs modulated in the same direction (both up or both down) were analyzed in relation to the microarray dataset (*ins* mutant vs phenotypically wild-type 108 hpf larvae). (F) Glucose levels measured in adult *Tg(ins:NTR)* animals after β-cell ablation and intraperitoneal injection with vehicle (DMSO) or flutamide; mean ± SEM, n = 6–8 animals. *P* values from t-tests.

DOI: https://doi.org/10.7554/eLife.42209.009

The following figure supplement is available for figure 4:

**Figure supplement 1.** Flutamide reduces glucose levels in a dose-dependent manner, possibly exerting its effects through liver gluconeogenic enzymes.

DOI: https://doi.org/10.7554/eLife.42209.010

(*Navarro et al., 2015*). Under high fat diet, a combination of neuronal and pancreatic β-cell specific roles for AR have been proposed to predispose female mice with androgen excess to diabetes (*Navarro et al., 2018*). Supporting this role, *ar* gene expression was observed in the zebrafish central nervous system and, additionally, in the liver (*Gorelick et al., 2008*) (*Figure 4—figure supplement 1B*). To investigate how AR antagonism mediates glucose level reduction in zebrafish *ins* mutants, we used a transcriptomic approach. RNA-Seq analyses of 120 hpf *ins* mutants treated with flutamide or cyproterone revealed 504 and 476 differentially expressed genes (DEGs) compared to vehicle treated mutants (*Figure 4D*), respectively. Of these DEGs, 40 were regulated in parallel (both up or both down) for both AR antagonists tested, likely highlighting the common AR-specific effects. Cross-referencing these 40 genes with a transcriptomic comparison of *ins* mutants to phenotypically wild-type siblings, led to 12 genes (*Figure 4E*) that were differentially expressed upon loss of *ins*, and were partially or fully restored to wild-type levels upon treatment with AR antagonists (*Figure 4—figure supplement 1C*). Amongst these 12 genes, *btg2* and *insig1* have been reported to play crucial roles in controlling liver gluconeogenesis (*Carobbio et al., 2013*; *Kim et al., 2014b*), and they also contain two androgen response elements (AREs) close to their transcription start site (*Figure 4—figure supplement 1D*). Additionally, upon intraperitoneal injections of flutamide in hyperglycemic adult animals (*Figure 4—figure supplement 1E*), we observed 19% lower fasting plasma glucose levels (*Figure 4F*), likely due to reduced hepatic glucose production. Our findings corroborate the observations of better anthropometric indices previously observed with flutamide (*Sahin et al., 2004*) or metformin +flutamide combination therapies (*Gambineri et al., 2004*; *Amiri et al., 2014*), and attribute a part of this beneficial effect to flutamide's insulin-independent action through AR antagonism.

In conclusion, to the best of our knowledge, ours is the first study to report the generation and use of a rapid screening strategy to identify insulin-independent pathways modulating metabolism in vertebrates. Given the recent success of SGLT2 inhibitors as combination therapy in diabetes (*Bailey et al., 2013*), our study is an important step towards identifying more insulin-independent mechanisms governing glucose homeostasis. One of the limitations of our screen is the relatively low size of the chemical library screened. However, as the endoderm transplant technique reported here can be combined with several genetic or metabolic readouts, future studies with larger chemical libraries should unveil mechanisms governing other disease-relevant phenomena as well. Such comprehensive insight into insulin-independent mechanisms and their interactions with insulin signaling in homeostasis and disease will open new avenues for designing therapies to treat metabolic disorders.

# Materials and methods

## Key resources table

| Reagent type or resource | Designation | Source or reference | Identifiers | Additional information |
|---|---|---|---|---|
| Genetic reagent (*Danio rerio*) | *ins^bns102* | This paper | | 16 bp deletion allele of *ins* (Gene ID: 30262) |
| Antibody | α-Insulin (Guinea Pig Polyclonal) | Dako | A0564 | (1:300) |
| Sequence-based reagent | *sox32* | *Kikuchi et al., 2001* | | (RNA) |
| Commercial assay or kit | mMessage mMachine SP6 Transcription Kit | ThermoFisher | AM1340 | |
| Commercial assay or kit | Glucose assay kit | Merck | CBA086 | |
| Software, algorithm | ZEN Blue 2012 | Zeiss, Germany | | |
| Software, algorithm | GraphPad Prism 7 | GraphPad Software, California | | |

## Zebrafish lines

Zebrafish husbandry was performed under standard conditions in accordance with institutional (MPG) and national ethical and animal welfare guidelines. The transgenic and mutant lines used in this study are *Tg(ins:DsRed)*$^{m1018}$ (*Anderson et al., 2009*), *Tg(-4.0ins:GFP)*$^{zf5}$ (*Huang et al., 2001*), *Tg(sox17:GFP)*$^{s870}$ (*Sakaguchi et al., 2006*), *Tg(ins:Flag-NTR, cryaa:mCherry)*$^{s950}$ (*Andersson et al., 2012*), *Tg(ins:EGFP-HRas, cryaa:mCherry)*$^{bns294}$, *Tg(ins:TagRFPt-P2A-insB)*$^{bns285}$, *ins*$^{bns102}$ (*ins* mutants), and *insb*$^{bns295}$ (*insb* mutants).

1, 2, 3, 4, 5, and 6 days post fertilization (dpf) correspond to 24, 48, 72, 96, 120, and 144 hr post fertilization (hpf), and 3 months post fertilization (mpf) corresponds to 90 dpf.

## CRISPR/Cas9 mutagenesis

CRISPR design platform (http://crispr.mit.edu) was used to design sgRNAs against *ins* (targeting sequence: TCCAGTGTAAGCACTAACCCAGG) and *insb* (targeting sequence: GGATCGCAGTCTTC TCC) genes and constructs were assembled as described previously (*Jao et al., 2013*; *Varshney et al., 2015*). Briefly, a mixture of 25 pg gRNA with 300 pg *Cas9* mRNA was injected into one cell stage wild-type embryos. High-resolution melt analysis (HRMA) (Eco-Illumina) was used to determine efficiency of sgRNAs and genotype animals with *ins* primers 5'-GTGCTCTGTTGGTCCTG TTGG-3' and 5'-CATCGACCAGATGAGATCCACAC-3', and *insb* primers: 5'-AGTATTAATCCTGC TGCTGGCG-3'and 5'-GTGTAGAAGAAACCTCTAGGC-3'.

## Immunostaining and Nile Red staining

Immunostaining and imaging was performed as described previously (*Yang et al., 2018*). Briefly, zebrafish larvae were euthanized and fixed overnight at 4°C with 4% paraformaldehyde (dissolved in buffer with composition: 22.6 mM $NaH_2PO_4$, 77 mM $Na_2HPO_4$, 120 µM $CaCl_2$, 117 mM sucrose, pH 7.35). After two PBS washes, the larvae were deskinned, and permeabilized using PBS containing 0.5% TritonX-100% and 1% DMSO for 1 hr. Larvae were then incubated in blocking buffer (Dako) containing 5% goat serum for 2 hr, and incubated with primary antibody overnight at 4°C. Next, samples were washed 3 × 10 min with PBS containing 0.1% TritonX-100, incubated overnight at 4°C with secondary antibody and DAPI (10 µg/ml), washed 3 × 10 min and mounted in agarose. Antibody dilutions used are as follows: guinea pig anti-Insulin polyclonal (1:100, Thermo), mouse anti-Glucagon (1:300, Sigma), chicken anti-GFP (1:300, Aves), goat anti-guinea pig AlexaFluor568 (1:300, Thermo), goat anti-mouse AlexaFluor647 (1:300, Thermo), goat anti-chicken AlexaFluor488 (1:500, Thermo). Zeiss LSM700 (10X) and LSM800 (25X) were used to acquire data, and Imaris (Bitplane) was used to analyze data and to create maximum intensity projection images.

Neutral lipid staining using Nile Red dye was performed at a working concentration of 0.5 µg/mL for 30 min in the dark, followed by acquisition of fluorescent images using an LP490 filter on a Nikon SMZ25 stereomicroscope.

## Morpholino injections

For knockdown of gene expression, the following splice-blocking antisense morpholinos (Gene Tools, LLC) were injected into one-cell embryos at the indicated dosage per embryo: *insa* MO (4 ng, 5'-CCTCTACTTGACTTTCTTACCCAGA-3') (*Ye et al., 2016*) *ar* MO (1 ng, 5'-AGCAGAGCCGCCTC TTACCTGCCAT-3') (*Peal et al., 2011*) standard control MO (4 or 1 ng, 5'-CCTCTTACCTCAGTTA-CAATTTATA-3').

## Intraperitoneal injections

Intraperitoneal injections and glucose level measurement in 6-month-old adult zebrafish was performed as described previously (*Curado et al., 2008*; *Moss et al., 2009*; *Eames et al., 2010*). Briefly, ablation of β-cells in *Tg(ins:Flag-NTR, cryaa:mCherry)*$^{s950}$ (*Andersson et al., 2012*) animals was performed by injecting 0.25 gm MTZ/kg body weight twice – on day 0 and day 1 – injecting twice improved the consistency of ablation. Flutamide (10 mg/kg) or vehicle (DMSO) was injected on days 2, 3 and 4. For injections, animals were anaesthetized using 0.02% Tricaine. On day 4, animals were euthanized and blood glucose was measured using a FreeStyle Freedom Lite Glucose Meter (Abbott).

## Larval glucose measurement

Free glucose level measurements were performed as described previously (*Jurczyk et al., 2011*), with minor modifications. After desired treatment conditions, pools of 10 animals were collected in 1.5 mL Eppendorf tubes and frozen at −80°C after complete removal of water. For analysis, pools of wild-type embryos were resuspended in PBS. Samples were homogenized using a tissue homogenizer (Bullet Blender Gold, Next Advance). A Glucose Assay Kit (CBA086, Merck) was used for glucose detection. Different volumes were used for resuspension and glucose detection between wild types and *ins* mutants: wild-type samples were resuspended in a volume corresponding to 8 µl/animal and 8 µl was used for the glucose detection reaction. *ins* mutant embryos, due to their much higher glucose content, were resuspended in a volume corresponding to 16 µl/animal and only 2 µl was used for glucose detection.

## Transplantations

For the endoderm transplant experiment, *sox32* mRNA was transcribed using an Sp6 mMessage mMachine kit (Ambion). Using a micro-injector, 100 pg of *sox32* mRNA was injected into *Tg(ins: DsRed)* embryos, which served as donors. Embryos from an *ins* +/- incross served as hosts. Between the 1 k-cell and sphere stages (3–4 hpf), 15–20 cells from donor embryos were transplanted to host embryos, targeting the host mesendoderm at the margin of the blastoderm.

## Wholemount in situ hybridization

Larvae were collected at 120 hpf and fixed with 4% paraformaldehyde in PBS overnight at 4°C. In situ hybridization was performed as described previously (*Thisse and Thisse, 2008*). *ar* digoxigenin-labelled anti-sense probe was synthesized using T7 polymerase (Roche) and DIG RNA labelling kit (Roche). The probe template was amplified using the following primers: *ar* ISH-forward 5'-TGGAG TTTTTCCTTCCTCCA-3' and *ar* ISH-reverse 5'- TAATACGACTCACTATAGGGTCATTTGTGGAA-CAGGATT- 3', obtaining a 1100 bp probe as described previously (*Gorelick et al., 2008*). Embryos were imaged on a Nikon SMZ25 stereomicroscope. Wild-type and mutant larvae were processed in the same tube and genotyped after the images were taken.

## Drug screening

3-mercaptopicolinic acid (3 MPA) treatment was performed at 1.5 mM concentration, metformin treatment at 250 µM concentration, and enzalutamide and bicalutamide treatments at 20 µM concentration. All other drug treatments were performed at 10 µM. For plate based screening, three 84 hpf *ins* mutant larvae were placed in each well of a 96-well plate in 200 µl of egg water buffered with 10 mM HEPES. All drug treatments were performed at 10 µM with 1% DMSO, unless otherwise stated. Drug treatment was performed from 84 to 120 hpf, after which each well was visually analyzed to assess toxicity. Subsequently, 100 µl of egg water was removed and 25 µl of 5X cell culture lysis buffer (Promega) was added. The plate was left shaking for 1 min at 750 rpm, and after gentle shaking at 150 rpm for 30 min, another round of vigorous shaking was performed for 1 min at 750 rpm. 8 µl from each well was used for the glucose detection reaction in a new 96-well plate using the Glucose Assay Kit (CBA086, Merck).

Drug libraries used in this screen are:

1. 1440 molecules from Edelris Keymical Collection (Edelris) (0 hits).
2. 285 molecules from the CLOUD collection (*Licciardello et al., 2017*) (two hits).
3. 156 molecules identified as *insulin* stimulators (*Matsuda et al., 2018*) (one hit).
4. 352 kinase inhibitors (SelleckChem) (0 hits)

## Transcriptomic analyses

For RNA-seq analysis, total RNA was isolated from 120 hpf zebrafish using the RNA Clean and Concentrator kit (Zymo Research), and samples were treated with DNase (RNase-free DNase Set, Promega) to avoid contamination by genomic DNA. Integrities of the isolated RNA and library preparation were verified with LabChip Gx Touch 24 (Perkin Elmer). 3 µg of total RNA was used as input for Truseq Stranded mRNA Library preparation following manufacturer's 'low sample' protocol (Illumina). Sequencing was performed on NextSeq500 instrument (Illumina) using v2 chemistry,

resulting in a minimum of 23M reads per library with $1 \times 75$ bp single end setup. The resulting raw reads were assessed for quality, adapter content and duplication rates with FastQC (*Andrews, 2010*). Trimmomatic (version 0.33) was employed to trim reads after a quality drop below a mean of Q20 in a window of 5 nucleotides (*Bolger et al., 2014*). Only reads between 30 and 150 nucleotides were cleared for further analyses. Trimmed and filtered reads were aligned with the Ensembl Zebrafish genome version DanRer10 (GRCz10.90), using STAR 2.4.0a with the parameter '– outFilterMismatchNoverLmax 0.1' to increase the maximum ratio of mismatches to mapped length to 10% (*Dobin et al., 2013*). The number of reads aligning to genes was counted with featureCounts 1.4.5-p1 tool from the Subread package (*Liao et al., 2014*). Only the reads that mapped, at least partially, to within exons were admitted and aggregated for each gene. Reads that overlapped multiple genes or aligned to multiple regions were excluded. Differentially expressed genes were identified using DESeq2 version 1.62 (*Love et al., 2014*). Maximum Benjamini-Hochberg corrected p-value of 0.05, along with a minimum combined mean of 5 reads, were set as inclusion criteria. The Ensembl annotation was enriched with UniProt data (release 06.06.2014) based on Ensembl gene identifiers ('Activities at the Universal Protein Resource (UniProt)," 2014). RNA-seq data have been deposited in the ArrayExpress database at EMBL-EBI (www.ebi.ac.uk/arrayexpress) under accession number E-MTAB-7283. From this dataset, normalized read counts for *ins* and *insb* expression are:

| Condition | *ins* | *insb* |
|---|---|---|
| DMSO | 183 | 1 |
| Flutamide | 176 | 2 |
| Cyproterone | 173 | 2 |

Average normalized counts for *ins* and *insb* from transcriptomic data from zebrafish adult β-cells (*Tarifeño-Saldivia et al., 2017*) are 4324351 and 0 respectively.

For microarray expression profiling, RNA was isolated from pooled 108 hpf zebrafish larvae using the RNA Clean and Concentrator kit (Zymo Research) combined with DNase digestion (RNase-free DNase Set, Promega). 10 animals were used for each pooled sample. Sample quality was tested using a Bioanalyzer and microarray analysis was performed by Oak Labs (Germany). Microarray data have been deposited in the ArrayExpress database at EMBL-EBI (www.ebi.ac.uk/arrayexpress) under accession number E-MTAB-7282.

## Mass spectrometric analysis

For each of the three biological replicates within a genotype, protein was extracted from pools of 600 larvae at 5 dpf using 4% SDS in 0.1 M Tris/HCl, pH 7.6 and a tissue disrupting sterile pestle (Axygen) for lysis. After heating to 70°C at 800 rpm for 10 min and DNA shearing by sonication, cell debris was removed by centrifugation at 14.000 x g for 10 min and retaining the supernatant. Using a DC protein assay (BioRad), 7 mg of solubilized proteins per sample were acetone precipitated at −20°C overnight, followed by centrifugation at 14.000 x g for 10 min and washing the pellet using 90% acetone. After evaporation of residual acetone, samples were dissolved in urea buffer (6 M urea, 2 M thiourea, 10 mM HEPES, pH 8.0), followed by enzymatic peptidolysis as described (*Graumann et al., 2008*; *Kim et al., 2018*) with the following modifications: 10 mM dithiothreitol, 55 mM iodoacetamide and 100:1 protein to enzyme ratio of the proteolytic enzymes were used. Subsequent sample processing and data analyses were performed as described previously (*Sokol et al., 2018*). The mass spectrometry proteomics data have been deposited to the ProteomeXchange Consortium via the PRIDE partner repository with the dataset identifier PXD012027.

Canonical pathway analysis was performed using Ingenuity Pathway Analysis (IPA) (Qiagen). Differentially expressed proteins from our study (log2FC ± 1.5) and from previously published datasets were subjected to a Comparison analysis in IPA. *P*-value maximum cut-off was set at 0.05 and the processes are listed according to those affected across most studies.

## Acknowledgements

We would like to thank Drs. Rashmi Priya and Sonja Sievers for discussion and comments, Dr. Radhan Ramadass for expert help with microscopy, Jasmin Gäbges and Roshan Satapathy for

experimental assistance, and Martin Laszczyk and the rest of the animal caretaking team for zebra-fish care. These studies were supported by funding from the Max Planck Society (DYRS).

## Additional information

### Competing interests
Didier YR Stainier: Senior editor, *eLife*. The other authors declare that no competing interests exist.

### Funding

| Funder | Grant reference number | Author |
|---|---|---|
| Max-Planck-Gesellschaft | Open-access funding | Sri Teja Mullapudi<br>Christian SM Helker<br>Giulia LM Boezio<br>Hans-Martin Maischein<br>Anna M Sokol<br>Stefan Guenther<br>Hiroki Matsuda<br>Johannes Graumann<br>Yu Hsuan Carol Yang<br>Didier YR Stainier |
| Deutsches Zentrumfür Herz-Kreislaufforschung | | Anna M Sokol<br>Johannes Graumann |

The funders had no role in study design, data collection and interpretation, or the decision to submit the work for publication.

### Author contributions
Sri Teja Mullapudi, Conceptualization, Data curation, Formal analysis, Investigation, Visualization, Methodology, Writing—original draft, Project administration, Writing—review and editing; Christian SM Helker, Resources, Supervision, Validation, Methodology, Writing—review and editing; Giulia LM Boezio, Validation, Investigation, Writing—review and editing; Hans-Martin Maischein, Investigation, Methodology; Anna M Sokol, Stefan Guenther, Johannes Graumann, Resources, Software, Formal analysis, Investigation, Writing—review and editing; Hiroki Matsuda, Resources, Writing—review and editing; Stefan Kubicek, Resources, Writing—review and editing, Designed the CLOUD drug library; Yu Hsuan Carol Yang, Conceptualization, Resources, Supervision, Methodology, Writing—review and editing; Didier YR Stainier, Conceptualization, Resources, Supervision, Funding acquisition, Validation, Writing—original draft, Project administration, Writing—review and editing

### Author ORCIDs
Sri Teja Mullapudi (iD) http://orcid.org/0000-0002-3916-8148
Hiroki Matsuda (iD) http://orcid.org/0000-0001-8639-2719
Stefan Kubicek (iD) https://orcid.org/0000-0003-0855-8343
Didier YR Stainier (iD) http://orcid.org/0000-0002-0382-0026

### Ethics
Animal experimentation: Zebrafish husbandry was performed under standard conditions in accordance with institutional (MPG) and national ethical and animal welfare guidelines approved by the ethics committee for animal experiments at the Regierungspräsidium Darmstadt, Germany (permit numbers B2/1017, B2/1041, B2/1089, B2/1138 and B2/Anz. 1007).

### Decision letter and Author response
Decision letter https://doi.org/10.7554/eLife.42209.020
Author response https://doi.org/10.7554/eLife.42209.021

# Additional files

## Supplementary files

• Supplementary file 1. List of proteins with Log2FC > 1 or Log2FC < −1 from proteomic analyses comparing 120 hpf *ins* mutant and wild-type animals.
DOI: https://doi.org/10.7554/eLife.42209.011

• Transparent reporting form
DOI: https://doi.org/10.7554/eLife.42209.012

## Data availability

Microarray and RNA-Seq data have been deposited in the ArrayExpress database at EMBL-EBI (www.ebi.ac.uk/arrayexpress) under accession number E-MTAB-7282 and E-MTAB-7283 respectively. Proteomics data has been deposited in the PRIDE database (https://www.ebi.ac.uk/pride/archive/) under accession number PXD012027.

The following datasets were generated:

| Author(s) | Year | Dataset title | Dataset URL | Database and Identifier |
|---|---|---|---|---|
| Sri Teja Mullapudi | 2018 | Microarray analysis comparing zebrafish insulin mutants to non-mutants | https://www.ebi.ac.uk/arrayexpress/experiments/E-MTAB-7282 | ArrayExpress, E-MTAB-7282 |
| Sri Teja Mullapudi | 2018 | Effects of anti-androgenic compounds on zebrafish insulin mutants | https://www.ebi.ac.uk/arrayexpress/experiments/E-MTAB-7283 | ArrayExpress, E-MTAB-7283 |
| Anna M Sokol | 2018 | Screening for insulin-independent pathways that modulate glucose homeostasis identifies androgen receptor antagonists | https://www.ebi.ac.uk/pride/archive/projects/PXD012027 | PRIDE, PXD012027 |

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
