## [Decision Letter]

Thank you for submitting your article "Screening for insulin-independent pathways that modulate glucose homeostasis identifies androgen receptor antagonists" for consideration by *eLife*. Your article has been reviewed by two peer reviewers, and the evaluation has been overseen by a Reviewing Editor and Mark McCarthy as the Senior Editor. The following individual involved in review of your submission has agreed to reveal his identity: Charles Hong (Reviewer #3).

The reviewers have discussed the reviews with one another and the Reviewing Editor has drafted this decision to help you prepare a revised submission.

Overall, the reviewers were enthusiastic about this manuscript and think that it adds an important new tool to the glucose homeostasis field. It will be of great interest to both zebrafish and non-zebrafish researchers and imagine it can be easily extended to identify even more pathways. The requests from the reviewers are relatively minor.

– More commentary on the efficacy of known anti-diabetic drugs, as depicted in Figure 3A, would be helpful in arriving at an understanding of the model itself. Why do some drugs work and others not, and what does this tell us about the strengths/limitations of the model?

– Similarly, more commentary on the human effects of the three hits would be helpful. In particular, hyperglycemia is a well-known side effect of vorinostat, so discussion of why zebrafish mutants respond differently would be helpful.

– It would be interesting to know if the glucose lowering effects of known drugs or novel hits prolonged survival in the *ins* mutants.

– Because the *insb* gene is apparently still intact in the *ins* mutants (although presumably expressed only at a low level), it might be helpful to show that the effects reported as insulin independent are also independent of *insb*.

– In Figure 3C, it appears that there are many more than 3 compounds that lowered glucose levels in the 2 replicate experiments. I assume of these, only 3 were found to "reproducibly reduced glucose levels upon retesting…" Can the authors explicitly state the number of compounds that made their prespecified cut-off in 2 replicates (perhaps in the figure legend)? This assay will be of interest to many investigators, and it would be good to know what the hit confirmation rate is.

– In Figure 4F, it appears that there is a trend toward lower fasting glucose levels in "diabetic" adult fish treated with flutamide, but this does not reach statistical significance (p=0.069). Looking at the data, this is likely due to the fact that only 5 fish were in the DMSO group. A few additional diabetic adult fish treated with DMSO would enable authors to make a more definitive conclusion.

---

## [Author Response]

Overall, the reviewers were enthusiastic about this manuscript and think that it adds an important new tool to the glucose homeostasis field. It will be of great interest to both zebrafish and non-zebrafish researchers and imagine it can be easily extended to identify even more pathways. The requests from the reviewers are relatively minor.– More commentary on the efficacy of known anti-diabetic drugs, as depicted in Figure 3A, would be helpful in arriving at an understanding of the model itself. Why do some drugs work and others not, and what does this tell us about the strengths/limitations of the model?

We have incorporated additional points into the Results and Discussion section to discuss the implications of our results with the known anti-diabetic drugs. To provide greater detail, we have also included a Pck1 inhibitor, 3-MPA, in our analyses in Figure 3A.

– Similarly, more commentary on the human effects of the three hits would be helpful. In particular, hyperglycemia is a well-known side effect of vorinostat, so discussion of why zebrafish mutants respond differently would be helpful.

We have incorporated additional points into the Results and Discussion section to address this question.

– It would be interesting to know if the glucose lowering effects of known drugs or novel hits prolonged survival in the ins mutants.

Glucose lowering drugs did not prolong survival of the *ins* mutants.

Insulin plays a versatile role in growth and metabolic signaling during development (Taniguchi et al., 2006). While our study contributes to the search for a small molecule as potent as insulin, we do not expect that lowering glucose levels alone will normalize all the processes dysregulated in *ins* mutants. We have now incorporated this information into the Results and Discussion section.

– Because the insb gene is apparently still intact in the ins mutants (although presumably expressed only at a low level), it might be helpful to show that the effects reported as insulin independent are also independent of insb.

*insb* expression peaks at 6 hours post fertilization (hpf) and is drastically decreased by 120 hpf (Papasani et al., 2006) (Figure 1—figure supplement 1F), and it is undetectable in adult β-cells (Tarifeño-Saldivia et al., 2017) with 0 normalized counts, as opposed to over 4 million for *ins*.

Our RNA-Seq analyses on 120 hpf *ins* mutants show normalized read count for *insb* transcript to be 1, which is minimal compared to 181 for *ins*. Additionally, upon treatment with the AR antagonists, Flutamide and Cyproterone, the normalized read counts for *insb* remain at a low level of 2 units (ArrayExpress accession number: E-MTAB-7283). Together, these data suggest that it is unlikely that *insb* could be contributing to the metabolic effects we observe. We have now incorporated these data into the Materials and methods section, under ‘Transcriptomic analyses’.

– In Figure 3C, it appears that there are many more than 3 compounds that lowered glucose levels in the 2 replicate experiments. I assume of these, only 3 were found to "reproducibly reduced glucose levels upon retesting…" Can the authors explicitly state the number of compounds that made their prespecified cut-off in 2 replicates (perhaps in the figure legend)? This assay will be of interest to many investigators, and it would be good to know what the hit confirmation rate is.

This information has been added to the legend of Figure 3.

– In Figure 4F, it appears that there is a trend toward lower fasting glucose levels in "diabetic" adult fish treated with flutamide, but this does not reach statistical significance (p=0.069). Looking at the data, this is likely due to the fact that only 5 fish were in the DMSO group. A few additional diabetic adult fish treated with DMSO would enable authors to make a more definitive conclusion.

We have now performed the experiment with additional animals and modified Figure 4 accordingly.